

**Precipitation reconstruction based on tree-ring width over the**
**past 270 years in the central Lesser Khingan Mountains,**
**Northeast China**
Mingqi Li[1,2], Guofu Deng[1,2], Xuemei Shao[1,2], Zhi-Yong Yin[1,3]
1 Key Laboratory of Land Surface Pattern and Simulation, Institute of Geographic Sciences and
Natural Resources Research, Chinese Academy of Sciences, Beijing 100101, China
2 University of Chinese Academy of Sciences, Beijing 100049, China
3 Department of Environmental and Ocean Sciences, University of San Diego, San Diego, CA
92110, USA
Correspondence: Mingqi Li (limq@igsnrr.ac.cn)



**Abstract:** Inter-annual variations in precipitation play important roles in management of forest
ecosystems and agricultural production in Northeast China. This study presents a 270-year
precipitation reconstruction of winter to early growing season for the central Lesser Khingan
Mountains, Northeast China based on tree-ring width data from 99 tree-ring cores of *Pinus*
*koraiensis* Sieb. et Zucc. from two sampling sites near Yichun. The reconstruction explained 43.9%
of the variance in precipitation from the previous October to current June during the calibration
period 1956-2017. At the decadal scale, we identified four dry periods that occurred during AD
1748-1759, 1774-1786, 1881-1886 and 1918-1924, and four wet periods occurring during AD 1790-
1795, 1818-1824, 1852-1859 and 2008-2017, and the period AD 2008-2017 was the wettest in the
past 270 years. Power spectral analysis and wavelet analysis revealed cyclic patterns on the inter-
annual (2-3 years) and inter-decadal (~11 and ~32-60 years) timescales in the reconstructed series,
which may be associated with the large-scale circulation patterns such as the Arctic Oscillation and
North Atlantic Oscillation through their impacts on the Asian polar vortex intensity, as well as the
solar activity.
**Key words:** tree-ring, precipitation reconstruction, the Lesser Khingan Mountains, Northeast China,
power spectral analysis and wavelet analysis



## Introduction

Precipitation is one of the most important climate variables in the global climate system and affects

human society via its impacts on water resources, agricultural production, and ecosystems. In recent

years, extreme droughts and flooding events repeatedly occurred in many regions of the world,

which have brought heavy losses in economy and human life. However, the scarcity of long-term

instrumental climatic data and historic records in many regions impedes our understanding to the

spatiotemporal precipitation variability and hampers our ability to plan for future. Additionally,

unlike temperature variation displaying relatively persistent patterns over large regions,

precipitation tends to have strong spatial variability. Therefore, spatially explicit and long-term data

are essential for understanding the current variation patterns and trends in the historical and spatial

context, which is also important for both validation of climate models and integration and

comparison with other historical, archaeological, and proxy data (Cook et al., 2010).

Tree-ring based reconstructions play an important role in paleoclimate studies due to their

accurate dating, annual resolution, wide distribution and good replication (Briffa et al., 1990; Cook

et al., 2000; Scuderi, 1993; Lamarche, 1974; Jacoby et al., 1996; Hughes et al., 1984; Shao et al.,

2005). In China, many tree-ring-based paleoclimate reconstructions are available in different

regions, such as the Tibetan Plateau (Zhang et al., 2003; Liang et al., 2009), Xinjiang Province

(Chen et al., 2014), Helanshan Mountain (Liu, 2004), and Hengduan Mountain (Fan et al., 2008; Li

et al., 2017). In comparison with these regions, long-term tree-ring based paleoclimatic records are

still relatively sparse for eastern China overall, including Northeast China, mostly due to long

history of human activities that have removed most old-growth forests.

The Lesser Khingan Mountains in Northeast China (Fig. 1) extends over 450 km from south



to north, and 210 km from east to west, occupying a total area of $7.77 \times 10^4$ km². Elevation varies
mostly between 500 and 1000 m above sea level (a.s.l.), with its highest peak (Mt. Pingdingshan)
at 1429 m a.s.l. (The Compilation Committee of Heilongjiang Local Gazzets, 1998). Northeast
China is a major agricultural region of China, as well as a region with rich forest resources, whose
total production of grain was 20.26% of the national total in 2018 (Tang et al., 2019). A thorough
understanding of precipitation variation and its impact on tree growth has signification implications
on the management of wildlife ecosystems. In recent years, there has been a moderate drying trend
with increased drought risk for most of Northeast China (Huang et al., 2017; Wang et al., 2015; Zhai
et al., 2017). With warming temperature, this tendency may be further enhanced due to increased
potential evapotranspiration (Kong et al., 2014). In order to assess future risk levels of droughts in
this region, it is necessary to put recent variations of moisture conditions in the long-term historical
context. Although several climate reconstructions have been developed in this area (Yu et al., 2018b;
Chen et al., 2016; Liu et al., 2010; Zhang et al., 2018b; Zhang et al., 2014; Yin et al., 2009), only a
few of them were precipitation reconstructions. For example, Zhang et al. (2014) reconstructed
previous-August to current July precipitation for Mohe of the Greater Khingan Mountains, while at
a larger regional scale, precipitation was reconstructed for southern Northeast China and the
northern Korean Peninsula (Chen et al., 2016). As stated earlier, since greater spatial variability is
seen in precipitation data than in temperature, there is the need to enhance spatial coverage of
precipitation reconstruction in this area.

<Insert Fig. 1 here>

Therefore, the goal of this study is to reconstruct the precipitation record based on tree-ring

width standard (TRW) chronology   from the central Lesser Khingan Mountains in Heilongjiang





Province of Northeast China. We hypothesize that moisture conditions during the early growing
season may serve as the control factor of radial growth of trees. Because of the dry winter and spring
seasons of the East Asian monsoon climate, late-spring/early-summer moisture conditions may
determine the pace of tree growth for the entire growing season to a large extent in this region.

## Materials and methods


### Study area and TRW chronology development


The study area is situated in Wuying District, Yichun City, Heilongjiang Province of Northeast
China in the central Lesser Khingan Mountains (Fig. 1). The topography of this area is characterized
by gentle hills with local relief of 285-688 m. The zonal soil is temperate dark brown soil, with
depth of 20-50 cm, developed on granite. There are many rivers, including Tangwang River, Fenglin
River, Pingyuan River, and nine other rivers, and the snowmelt water and precipitation in summer
are the supply of the rivers. The zonal vegetation is the conifer-broadleaf forest in this area, which
is one of the oldest virgin forests in the broadleaved-Korean pine forest ecosystem. Korean pine
(*Pinus koraiensis* Sieb. et Zucc), firs (*Abies fabri*), spruces (*Picea asperata*) and larch (*Larix gmelini*)
are main forest types, and Korean pine is the dominant primary forest species. We chose the mature
Korean pine forest for sampling on the hillside faced west and north-west in two sites (FL1 and
WY1), with dense canopy coverage (85%) and no signs of extensive logging activities. The distance
between two sites is about 7.5 km. Site information, including latitude and longitude, slope, aspect,
tree species and core/tree number, is listed for two sites in Table 1. The climate is influenced by the
East Asia monsoon and Siberian High system, belong to temperate continental climate with long
winters but warm, transitory summers (Zhao, 1995). There are two nearby meteorological stations,
Yichun and Tieli, which recorded a 1958-2017 mean annual temperature of 1.49℃, with a mean



temperature of -22.5℃ in January (the coldest month) and 21.3℃ in July (the warmest month).
Mean annual precipitation is 539.4 mm with approximately 84.6% occurring during May to
September (Fig. 2). In addition,some studies indicated that the abnormal climate in Heilongjiang
province in the early summer is related to the Asian polar vortex (Zhang and Li, 2013). It is also
found that the polar vortex intensity in December or winter is a factor on the precipitation in
Northeast China in the subsequent August or summer (Yao and Dong, 2000). Therefore, the Asian
polar vortex may be one of the factors influencing the precipitation in our study area.

<Insert Fig. 2 here>

<Insert Table 1 here>


We conducted field campaign in September, 2013 and 2017, and collected a total of 103 cores

from 53 living Korean pine from two sites using 10 mm diameter increment borers (Fig. 1 and Table
1). Annual ring widths were measured to a precision of 0.01mm using the LINTAB 6 ring-width
measurement system. The program COFECHA was used to test the accuracies of cross-dating and
measurement of ring widths (Holmes, 1983). Each individual ring-width series was fit to the
negative exponential or Hugershoff curve in order to remove non-climatic trends due to age, size,
and stand dynamics (Fritts, 1976; Cook et al., 1995). Standardization was performed using the
ARSTAN program (Cook, 1985). The detrended data from individual tree cores were combined into
site chronologies using a bi-weight robust mean (Cook and Kairiukstis, 1990), which minimizes the
influence of outliers (i.e., abnormal narrow and wide rings caused by certain factors other than
climate), extreme values, or biases in the tree-ring indices (Cook et al., 1990a). The ARSTAN
program produces three versions of standardized chronologies: Residual, Standard, and ARSTAN



and the Standard version was used in the following analysis.
The signal-to-noise ratio (SNR) was used to evaluate the relative strength of the common
variance signal in the tree-ring chronology (Wigley et al., 1984). The expressed population signal
(EPS) was calculated using a 50-year window with 25-year increments over the total length of the
series (Wigley et al., 1984). The EPS denotes the representativeness of a sample to the entire
population as a measure of signal quality, with values above 0.85 generally regarded as satisfactory
for dendroclimatic studies (Wigley et al., 1984).
**Meteorological and circulation data**
Climatic data records at the Yichun meteorological station (128.92˚E, 47.73˚N; 240.9 m a.s.l.,
Fig. 1) were compared herein to the TRW chronology, including monthly total precipitation (PPT),
monthly mean maximum temperature (TMAX), monthly mean temperature (TMEAN) and monthly
mean minimum temperature (TMIN) during 1956-2017. We also considered the possible lagged
effects of weather conditions on tree growth. We also collected the monthly Standardized
Precipitation-Evapotranspiration Index (SPEI) during the period of 1956-2013
(http://climatedataguide.ucar.edu/cliamte-data/standardized-precipitation-evapotranspiration-
index-spei) to calculate spatial correlations with the TRW chronology, and the gridded CRU TS 4.02
precipitation data for the period 1956-2017 (www.cru.uea.ac.uk) to further explore the spatial
representativeness of the reconstructed precipitation.
In addition, in order to discuss the possible driving factors that affected the precipitation regime,
we collected the Asian polar vortex intensity (APVI) data, a measure determined by the total air
mass quantity or density between 500 hPa geopotential height field and the isohypsic surface located
that the polar vortex southern boundary characteristic contour covering 60-150°E in Northern





Hemisphere, and these data were obtained from the website (https://www.ncc-
cma.net/Website/index.php?ChannelID=43&WCHID=5). We also collected large-scale circulation
patterns data that are known to have influence on weather conditions in China: El Niño/Southern
Oscillation (ENSO) (Trenberth and Stepaniak, 2001; Wu et al., 2003), Multivariate ENSO Index
(MEI, Wolter and Timlin, 1998) and Southern Oscillation Index (SOI, Troup, 1965), Pacific Decadal
Oscillation (PDO) (Mantua et al., 1997; Wang et al., 2008), Arctic Oscillation (AO) (Wu and Wang,
2002; Zhou et al., 2001; Thompson and Wallace, 1998), and North Atlantic Oscillation (NAO)
(Jones et al., 1997; Hurrell, 1995; Yao et al., 2017).
**Radial growth - climate relationships, reconstruction calibration and verification**
To investigate the tree growth-climate relationships, we calculated the Pearson's correlation
coefficients between the TRW chronology and TMEAN, TMAX, TMIN and PPT during the
instrumental period of 1956-2017. Since the climate of a given year could have a lagged effect on
the growth in the following year (Fritts, 1976), climate data from the previous October to the current
September were used in the correlation analysis. To test whether the correlation coefficients were
affected by variations in the low-frequency domain, we also calculated the correlation coefficients
using the first-differences of the chronology and the climatic data. The results can give us hints on
which climate variable served as the major limiting factor of radial growth of trees, the potential
target for reconstruction.
In reconstruction, we first established a transfer function using linear regression in which the
TRW chronology was used as the independent variables and the selected climatic factor as the
dependent variable for the full calibration period. To validate the transfer function, the cross-
validation procedure (Michaelsen, 1987) and independent split-period validation procedure (Fritts,



1976) were used in this study. The validation statistics include the sign tests on both the original and
first-difference data and t test of product means to show how well the model-predicted values
following the directions of variation in the observed values (Fritts, 1976). Also included are
reduction of error (RE), coefficient of efficiency (CE) and correlation coefficient. RE is a measure
of comparison between the predicted and observed values (Fritts, 1976), and CE is a relative
measure of the analysis error variance to the variance in the true state (Nash and Sutcliffe, 1970;
Tardif et al., 2014). Positive RE and CE values are evidence for a valid transfer function (Fritts,
1976; Nash and Sutcliffe, 1970).
**Power spectral analysis and wavelet analysis**
Spectral analysis is the process of estimating the power spectrum of a signal from its time-
domain representation. To examine the temporal variation pattern of precipitation in our study area
in different frequency domain, we performed power spectral analysis (Fowler, 2010) and wavelet
analysis (Torrence and Compo, 1998).
# Results
## Characteristics of the TRW chronology
The two sites are very close, and the correlation coefficients between each series and master
dating series of flagged 50-year segments (lagged 25-year) filtered with 32-year spline were 0.61-
0.80 calculated using the COFECHA software. Therefore, we combined the tree-ring width data
when developing the TRW chronology. The TRW chronology covered the periods AD 1685-2017
(Fig. 3). The statistical characteristics of the chronology are given in Table 2. The mean sensitivity
(a measure of the inter-annual variability in tree-ring series) was 0.223, indicating that the TRW
chronology showed relatively low inter-annual variability compared to those chronologies from





semi-arid area (Shao et al., 2010). The first-order autocorrelation of the TRW series was 0.31,
suggesting that the radial growth was probably influenced by conditions of previous years. The Rbar
(overall mean correlations between the sample series), Rbt (mean between-tree correlations), and
Rwt (mean within-tree correlations) were 0.258, 0.251 and 0.801, respectively. They were
comparable to other tree ring studies in the region (e.g., Yin et al., 2009). Beginning in 1748, the
chronology can be considered reliable with sufficient numbers of samples as the EPS reached 0.85
with 17 cores. In addition, the SNR was 30.215. All statistics indicated that the chronology was
suitable for dendroclimatic reconstruction.

<Insert Fig. 3 here>

<Insert Table 2 here>

## Tree growth-climate relationships

Fig. 4 shows the results of correlation analysis of the TRW series with monthly TMEAN,

TMAX, TMIN and PPT. For the original data, positive correlations were found between temperature
and the TRW chronology from previous October to current September except for current June with
TMEAN and TMAX. The correlations with TMIN were consistently higher than those of TMEAN
and TMAX, and statistically significant at the 0.05 level for previous October, current January-June,
and August. Positive correlations were also found between PPT and the TRW chronology from
previous October to current September except for current March and August, but only the correlation
coefficient in current June was statistically significant (Fig. 4A). After first-differencing of the data,
the positive correlation with June PPT still remained statistically significant, although weaker (Fig.
4B). In the meantime, the positive correlations with temperature variables from previous October to
current May became weaker, while the negative correlations from current June to September became



stronger, especially for June TMEAN and TMAX (Fig. 4B) indicating the effect of vegetation water
use stress associated with high temperatures during the growing season. The differences between
the results for the original and first-difference data suggest that the positive correlations between
the temperature variables and the TRW chronology were probably mostly resulted from variations
in the low-frequency domain, as they became weaker for the first-different data. However, the
signals of early growing season moisture conditions remained strong in the high-frequency domain,
as indicated by the persistent correlations with PPT and stronger negative correlations with TMEAN
and TMAX in June for the first-difference data (Fig. 4B). We also calculated the correlations
between the TRW chronology and climatic variables for different combinations of months/seasons.
The strongest correlation was produced using a combined variable of previous October-current June
total precipitation for the origin data (r=0.663, p<0.01), which was also statistically significant for
the first-difference data (r=0.438, p<0.01). These results suggest that the cold-season and early
growing-season precipitation is a major factor of radial growth of trees at our sampling site, with its
effects detectable in the TRW series variations in both low- and high-frequency domains.

<Insert Fig. 4 here>

## 222 **Calibration and verification of the transfer function for reconstruction**

Based on the growth-climate relationships during the period 1956-2017 (Fig. 4), we decided to
reconstruct the total precipitation from previous October to current June ($PPT_{p10-c6}$) using the TRW
chronology (Fig. 5A). Linear regression was used to calibrate the transfer function using data from
1956 to 2017:
$PPT_{p10-c6} = 110 + 149 \text{ TRW}.$

<Insert Fig. 5 here>



The model explained 43.9% ($R_{adj}{}^2$=43%) of the variance in $PPT_{p10\text{-}c6}$ for the full calibration
period (Table 3). The sign test is statistically significant at the 0.01 level for the original data, but it
was not significant for the first-difference data. The result indicated that the match between the
reconstructed and observed rainfall data was better in the low-frequency domain than that in the
high-frequency domain. The relatively high values of RE and product mean t indicated reasonable
skill in the reconstruction with a leave-one-out correlation coefficient of 0.63. The results of split-
period validation are also presented in Table 3. In the first split-period validation, the calibration
period was set to be 1956-1986, and validation period as 1987-2017. The calibration model
explained 21.4% of the variance in $PPT_{p10\text{-}c6}$. Results of the signs tests for the original data (ST) and
first-difference data (ST1) were not significant at the 95% confidence level, but the RE and CE
values are above zero and the t value of product mean is high, again suggesting reasonable skills for
reconstruction with a correlation coefficient of 0.729 for the original-reconstructed climate in the
verification period. For the second split-period validation, the period 1987-2017 was used for
calibration and 1956-1986 for validation. The model explained 53.2% of the variance in $PPT_{p10\text{-}c6}$.
The sign test of the original data reached the 95% confidence level, but the result of the first-
difference data was not statistically significant. The correlation coefficient, RE, and CE were lower
than those of the first split-period validation, but remained positive, and the product mean t value
remained high. The validation results suggested that the model was relatively robust with sufficient
skills of estimation. The reconstructed precipitation series derived from the model showed a good
agreement with the observed precipitation values during the calibration period (Fig. 5B).

<Insert Table 3 here>

**Temporal variation of the reconstructed precipitation**



The reconstruction period began in AD 1748 when the TRW series' EPS exceeded 0.85 (Table
2 and Fig. 3). Fig. 5C shows the reconstructed $PPT_{p10-c6}$ during period of 1748-2017. The
reconstructed precipitation revealed strong inter-annual, decadal variations providing a valuable
long series to evaluate the local climate variability. Here, we designates a value of 1σ (σ = 17.75
mm) above the mean as wet year ($PPT_{p10-c6}$>269.599 mm), 1σ below the mean as dry year ($PPT_{p10-c6}$
$_{c6}$<234.101 mm), and the remaining as normal year. According to this criterion, four dry periods that
occurred during AD 1748-1759, 1774-1786, 1881-1886 and 1918-1924 with AD 1774-1786 as the
driest, and four wet periods occurring during AD 1790-1795, 1818-1824, 1852-1859 and 2008-2017,
and the period AD 2008-2017 was the wettest in the past 270 years on the decadal scale.
**Discussion**
**Responses of radial growth to climate**
Based on the correlations between TRW indices and climatic factors, the total precipitation
from previous October to current June played a key role in regulating the radial growth of Korean
pine in our study area, which indicated that the total precipitation during periods before and during
the early growing season is the major factor affecting the growth of Korean pine. Similar results
about the climate-tree growth relationship were found in Northeast China (Chen et al., 2012; Liu et
al., 2009; Liu et al., 2010; Yu et al., 2018a; Zhang et al., 2014; Wang and Lv, 2012) and other regions,
especially in semi-arid Northwest China (Fang et al., 2013; Liang et al., 2009). We speculate that
one reason is the snow accumulated early in the season, which can insulate the soil and contribute
to keeping warm soil temperatures in winter and rapid water absorption by the roots in the following
spring and early growing season (Fritts, 1976). In addition, the combination of a positive correlation
between the TRW chronology and June precipitation and negative correlations with the mean and





maximum June temperatures is indicative of moisture stress as the limiting factor of tree growth in
our study area, which is also common in many sub-humid to semi-arid regions in North and
Northwest China (Shao et al., 2010; Liu et al., 2010; Liu et al., 2013; Liu et al., 2004; Sun and Liu,
2013; Chen et al., 2014). Furthermore, we also calculated the correlation coefficients between the
TRW    chronology    and    SPEI    (http://climatedataguide.ucar.edu/cliamte-data/standardized-
precipitation-evapotranspiration-index-spei) in June for the period 1956-2013 and plotted the results
using KNMI Climate Explorer (https://climexp.knmi.nl/). The correlation coefficients varied
between 0.4 and 0.6 over a region covering approximately 40-51˚N and 121-130˚E (p<10%) (Fig.
6A), displaying a similar correlations with that of the TRW chronology and the precipitation in June,
but weaker than those between the TRW chronology and the precipitation from previous October to
current June. These results also supported the conclusion that moisture is the major factor affecting
the growth of Korean pine at our study sites.

<Insert Fig. 6 here>

In this region, there have been more reconstructions of the past temperature than precipitation.

For example, even at a site very close to ours (also in the Wuying District), Yin et al. (2009)
reconstructed temperature variations of the previous October using the same tree species. A further
comparison revealed that they used the residual chronology rather than the standard chronology, and
also used climatic data from a different meteorological station (Wuying rather than Yichun). In their
study, the only month of statistically significant correlations was October of the previous year and
they did not conduct correlation analysis for the first-difference data. After first differencing in our
analysis, all positive correlations with temperature variables became statistically insignificant at the
0.95 confidence level. Therefore, we are confident that the growth-precipitation relationship as



displayed in Fig. 4 is more robust than the relationship between tree growth and temperature.
However, we also speculate that this relationship may have been strengthened due to the recent
warming, as indicated by the better results for the second period 1987-2017 in the split-period
validation process. Moisture condition as the limiting factor was suggested by several studies on
tree growth responses to climatic factors in this region. For example, Zhu et al. (2015) pointed out
that the warming after 1980 caused the response of Korean pine growth to PDSI from a negative
correlation to a positive correlation, suggesting a greater influence of moisture conditions on radial
growth in the more recent period. In the meantime, Liu et al. (2016) examined four sites in Northeast
China following a latitudinal gradient and concluded that tree growth at different latitudes may have
different responses to climatic variables. However, the effect of early growing-season moisture
stress was visible in their results of growth-climate correlation analysis for the sites north and south
of our study area. Using ring-width data from three species including Korean pine, Zhang et al.
(2018a) reconstructed the July normalized difference vegetation index (NDVI) series for the
southern Lesser Khingan Mountains and concluded that the low values of the reconstructed NDVI
series corresponded to the drought periods since the 1900s, linking the tree ring width data to the
moisture conditions.
**Comparisons with other precipitation/drought reconstructions and the**
**representativeness of the reconstructed precipitation**
To assess the reconstructed precipitation variation, the dry and wet periods of the reconstruction
were compared with the January-March streamflow of the upper Nenjiang River (Wang and Lv,
2012), previous June-July PDSI in the Northern Daxing'anling (Greater Khingan) Mountains (Yu
et al., 2018a), and previous October-current September precipitation in the Southern Northeast



China and the Northern Korean peninsula (Chen et al., 2016). The results showed that several wet
and dry periods of our reconstruction corresponded well with the other reconstructed precipitation
and PDSI series (Fig. 7), suggesting persistent large-scale weather conditions affecting the entire
Northeast China.

<Insert Fig. 7 here>

The 1920s drought was one of the most severe and well-documented natural hazards in the last
200 years in the semi-arid and arid areas of northern China (Liang et al., 2006). In the Wuying area,
the 1920s was a dry period with the driest year in 1920 (Fig. 7). For the entire 1920s, however, the
moisture conditions gradually recovered from the low. Based on gridded temperature and
precipitation data, Ma et al. (2005) analyzed the shift of dry/wet boundaries for different regions in
China during 1900-2000. They discovered that for Northeast China, there was a wetting trend during
the 1920s, with the boundary of the semi-arid and sub-humid regions shifting westward from 128°E
to 124°E, which was then reversed in the early 1930s (Ma and Fu, 2005). In the meantime, most
other regions in China experienced the peak drought conditions during the late 1920s and early
1930s (Liang et al. 2006). Therefore, most likely this severe drought did not reach our study region
where the 1920 drought was a separate event impacting various regions in Northeast China (Fig. 7).
To further explore the spatial representativeness of the reconstructed precipitation series, we
calculated correlation coefficients between the observed (Fig. 6B) and reconstructed (Fig. 6C)
$PPT_{p10-c6}$ data for the period 1956-2017 using the gridded CRU TS 4.02 dataset ([www.cru.uea.ac.uk](http://www.cru.uea.ac.uk))
and plotted the results using KNMI Climate Explorer (https://climexp.knmi.nl/). The reconstructed
$PPT_{p10-c6}$ correlated significantly with the gridded precipitation over a region covering
approximately 42-52˚N and 124-132˚E (r>0.5, p<10%) (Fig. 6C), displaying a similar spatial



structure of the correlations (although weaker) between the observed $PPT_{p10\text{-}c6}$ and the gridded
precipitation data (Fig. 6B). These results indicated that our precipitation reconstruction can capture
the occurrences of drought events in a large area in the northern part of Northeast China.

## 342     Possible driving mechanisms

To examine the temporal variation pattern of precipitation in the Wuying area in different

frequency domains, which may allow us to explore possible driving factors that affected the
precipitation regime, we performed power spectral analysis of the reconstruction series and
discovered semi-cyclic variations with periods of 2.2-3.2 years, 11 years, and 30 years (Fig. 8A).
Wavelet analysis also confirmed these results, showing cyclic periodicities of 2-3 years, ~11 years,
and ~30-64 years (Fig. 8B).

<Insert Fig. 8 here>

Since early growing season moisture condition is the limiting factor of radial growth of trees

and more than 60% of the observed $PPT_{p10\text{-}c6}$ occurs in May and June, explaining more than 71%
of the total variance in $PPT_{p10\text{-}c6}$, we will focus on the atmospheric processes that influence May-
June precipitation in the following. At this time of the year, previous studies indicated that
precipitation in this region is mostly caused by extratropical cyclonic activities that are impacted by
the Asian Polar Vortex Intensity (APVI) (Zhang and Li, 2013), The correlation between the $APVI_{p10\text{-}}$
$_6$ and the observed $PPT_{p10\text{-}c6}$ at Yichun was -0.275 (p = 0.033), while its correlation with the
reconstructed series was -0.243 (p = 0.051). We argue that the APVI in May and June ($APVI_{c56}$)
would have a significant impact on $PPT_{p10\text{-}c6}$. This was validated by the correlations of the $APVI_{c56}$
with the observed (r = -0.375, p = 0.002) and reconstructed (r = -0.269, p = 0.029) $PPT_{p10\text{-}c6}$ series.
Therefore, in the following, we will focus on the relationships between $APVI_{c56}$ and various large-





scale circulation patterns influencing on weather conditions in china, including El Niño/Southern
Oscillation (ENSO) (Trenberth and Stepaniak, 2001; Wu et al., 2003), Multivariate ENSO Index
(MEI, Wolter and Timlin, 1998) and Southern Oscillation Index (SOI, Troup, 1965), Pacific Decadal
Oscillation (PDO) (Mantua et al., 1997; Wang et al., 2008), Arctic Oscillation (AO) (Wu and Wang,
2002; Zhou et al., 2001; Thompson and Wallace, 1998), and North Atlantic Oscillation (NAO)
(Jones et al., 1997; Hurrell, 1995; Yao et al., 2017)..

Both the ENSO and PDO did not show any significant correlation with the $APVI_{c56}$ (Table 4).

However, AO and NAO showed significant positive correlations with $APVI_{c56}$ (Table 4). Since the
AO and NAO time series are highly correlated to each other (Ambaum et al., 2001), we further
analyzed the temporal variation patterns of a reconstructed monthly NAO series since 1659
(Luterbacher et al., 2002). The correlation coefficient between the reconstructed May-June NAO
and reconstructed $PPT_{p10-c6}$ was -0.118 (p = 0.061) for the common period 1748-2001, while the
correlation between the two series after 5-year smoothing was -0.229 (p=0.2) after adjusting degree
of freedom according to the formula calculated by Bretherton et al. (1999). On the decadal scale,
the inverse correlation between the reconstructed $NAO_{c56}$ and reconstructed $PPT_{p10-c6}$ exists (Fig.
9). Power spectral analysis of this NAO series showed statistically significant cyclic patterns of 2.7-
3.2 years and 50-60 years, which matched the periodicities in the reconstructed $PPT_{p10-c6}$ series (Fig.
8a). This specific reconstructed May-June NAO series did not show a 30-year cyclic pattern.
However, it existed in a multi-proxy NAO reconstruction by Trouet et al. (2009). Finally, the 11-
year cycle in the reconstructed series matched the 11-year sunspot cycle, probably due to its impact
on the Asian Polar vortex at the 300 hPa geopotential height (Angell, 1992). Overall, we identified
the Asian Polar Vortex as the possible regional control factor of winter-early summer precipitation



in our study region, while AO and NAO are the most likely large-scale circulation patterns that
influence the inter-annual variation of precipitation in the Lesser Khingan Mountains. Contrary to
some previous studies (Zhang et al., 2018c), ENSO and PDO were not found to be related to winter-
early growing season precipitation in our study area.

<Insert Table 4 here>

<Insert Fig. 9 here>

## Conclusion

In this study we reconstructed winter to early growing-season precipitation based on the ring-
width chronology of *Pinus koraiensis* Sieb. et Zucc. during AD 1748-2017 in the Lesser Khingan
Mountains, using a total of 99 sample cores from 50 trees. The study region is characterized by a
humid continental climate where most previous climatic reconstructions focused on temperature
variations. In the climate-growth relationship analysis, correlation analysis between the TRW
chronology and climatic factors revealed strong signals of the early growing-season moisture deficit
as the major control factor of radial growth of trees. The transfer function explained 43.9% of the
variance in previous October-current June precipitation for the calibration period 1956-2017. This
270-year precipitation reconstruction showed good spatial representation and revealed four dry
periods that occurred during AD 1748-1759, 1774-1786, 1881-1886 and 1918-1924, with AD 1774-
1786 as the driest. It also revealed four wet periods occurring during AD 1790-1795, 1818-1824,
1852-1859 and 2008-2017, and the period AD 2008-2017 was the wettest in the past 270 years on
the decadal scale. In addition, although 1920 was a dry year in our study area, the severe drought
that hit many regions in North China during the late 1920s most likely spared this region. The results
of power spectral analysis and wavelet analysis revealed cyclic patterns of 2.3-3.2 years, 11 years,



and 30-64 years in the reconstructed precipitation series, which matched those of a reconstructed
NAO series and the 11-year sunspot cycle. Our results suggest that the Asian Polar Vortex is
probably the regional control factor of the inter-annual variation of winter-early growing season
precipitation, while NAO and AO are the associated large-scale circulation patterns. Results from
our study indicated that even in a cold and relatively humid climate, moisture condition can still
serve as a control factor for radial growth of trees, which provides more opportunities for climatic
reconstructions of precipitation to enhance spatial coverage of sampling sites as precipitation tends
to have strong spatial variability. This may also have significant implications in forest and
ecosystems management and agricultural production.

**Data availability.** Correspondence and requests for data should be addressed to Mingqi Li
(limq@igsnrr.ac.cn).
**Author contributions.** This study was designed by all authors. ML, XS and ZY conducted field
sampling, performed data processing and analysis, and wrote the manuscript. GD implemented the
power spectral analysis and possible driving mechanisms analyses.
**Competing interests.** The authors declare that they have no conflict of interest.
**Acknowledgements.** We are grateful to Professors Xiaochun Wang, Zhenju Chen and Jian Yu for
their providing the reconstructed data comparing with our reconstructed precipitation.
**Financial support.** This research was supported by the National Key R&D Program of China on
Global Change (grant No. 2017YFA0603302), and University of San Diego (FRG #2017-18 and

425 2019-20).

**Review statement.** This paper was edited by *** and reviewed by two anonymous referees.

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



**Figure captions**

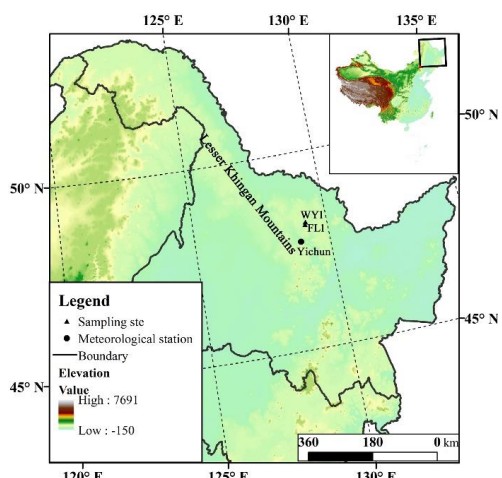


Fig. 1: Map showing locations of sampling sites and meteorological station.



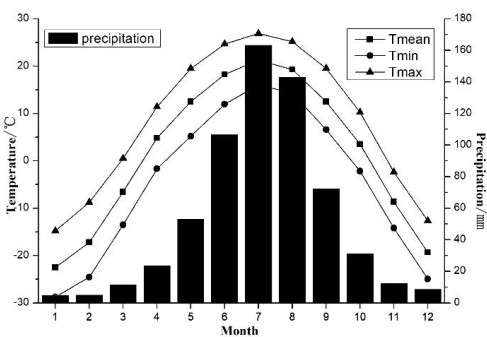

Fig. 2: Monthly mean temperature, maximum temperature, minimum temperature, and precipitation over the period 1958-2017 derived from meteorological station Yichun and Tieli.





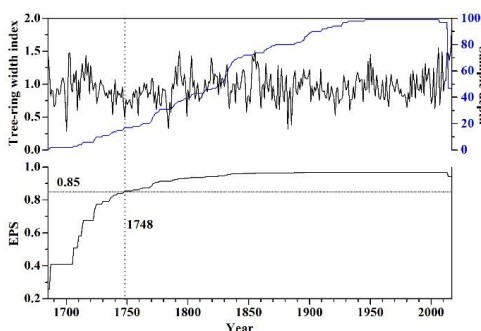

Fig. 3 the tree-ring width standard chronology, sample depth and EPS from the study site.





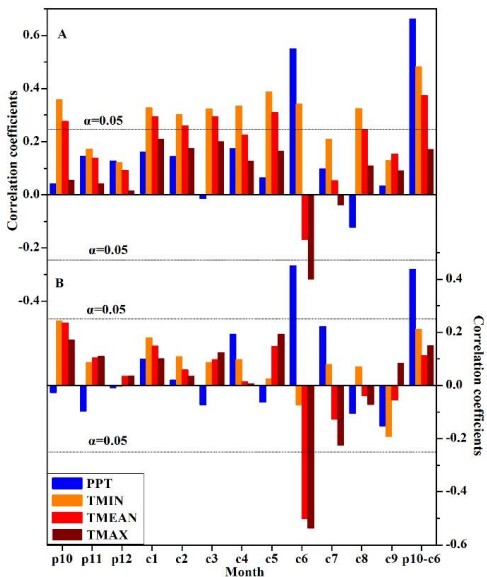

Fig. 4: Correlation coefficients between the TRW standard chronology and monthly temperature (TMIN, TMEAN, and TMAX) and precipitation (PPT) for the original (A) and first-difference (B) during 1956-2017.






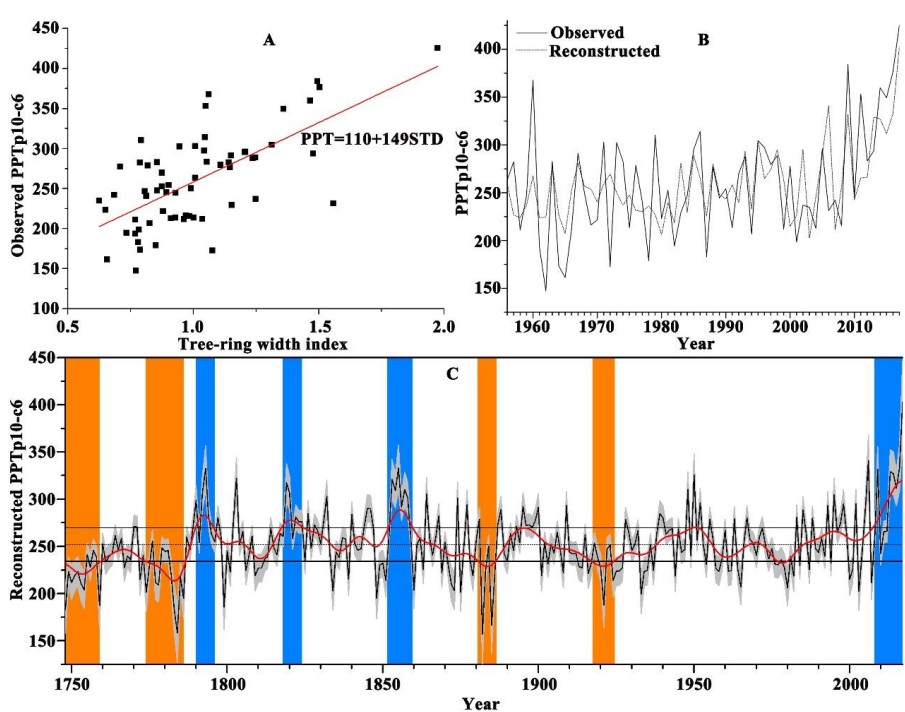


Fig 5: Scatter plot of the observed and tree-ring width index, regression line (red line) and equation (A); graph of the observed and reconstructed p10-c6 precipitation ($PPT_{p10-c6}$) for the full calibration period 1956-2017 (B); Reconstructed $PPT_{p10-c6}$ (black line) and 11-year smoothing (FFT filter) (red line), the gray area denotes the confidence interval at 95%, the orange area indicates the drought period, and the blue area is wet period (C).





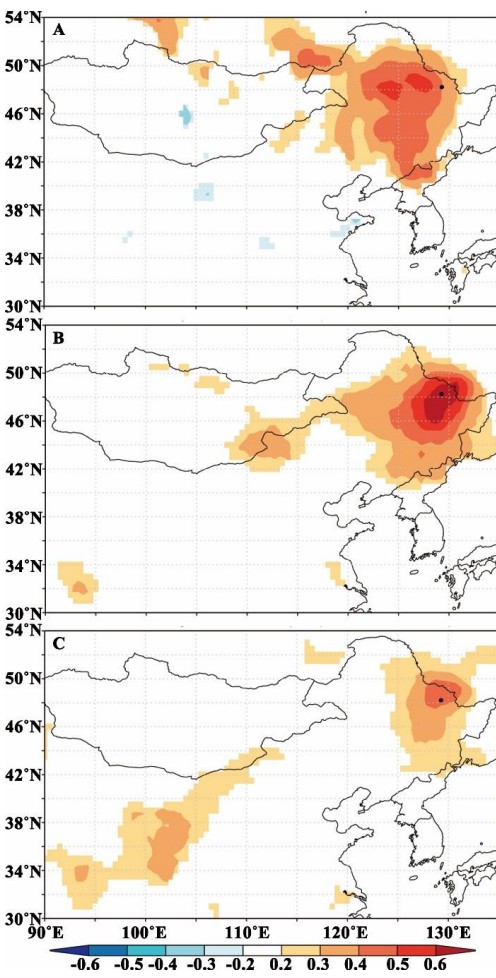


Fig. 6 Spatial correlation fields of the TRW chronology with the gridded SPEI
(http://climatedataguide.ucar.edu/cliamte-data/standardized-precipitation-evapotranspiration-
index-spei) on June for the period 1956-2013 (A, https://climexp.knmi.nl), and the observed (B) and
reconstructed (C) $PPT_{p10-c6}$ with the gridded CRU TS 4.02 precipitation (www.cru.uea.ac.uk) from
previous October to current June (https://climexp.knmi.nl) for the period 1956-2017. The black
circle dots are the our sampling sites.

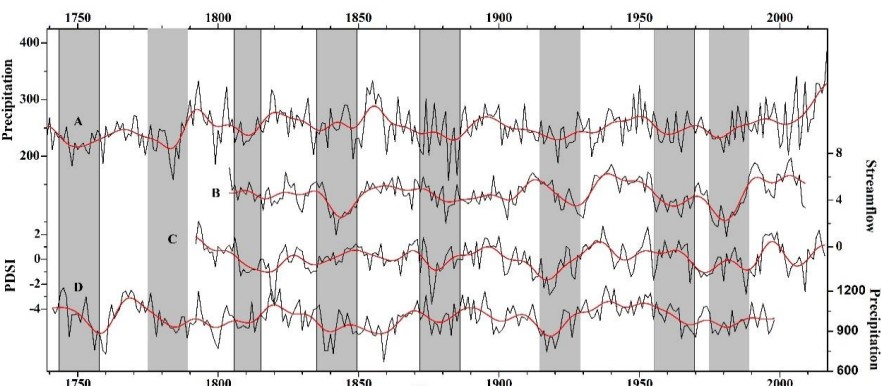

Fig. 7 Comparisons of the p10-c6 precipitation reconstruction with other tree-ring reconstructions in the northeastern China. The vertical shading indicated the periods of drought in the reconstructed precipitation series when 11-year smoothed values were lower than the long-term mean. (A) the reconstructed precipitation in this study; (B) the reconstructed streamflow of the upper of the Nenjiang River (Wang and Lv, 2012); (C) the reconstructed PDSI of Northern Daxing'anling Mountains (Yu et al., 2018a); (D) the reconstructed precipitation of southern Northeast China and the northern Korean peninsula (Chen et al., 2016).






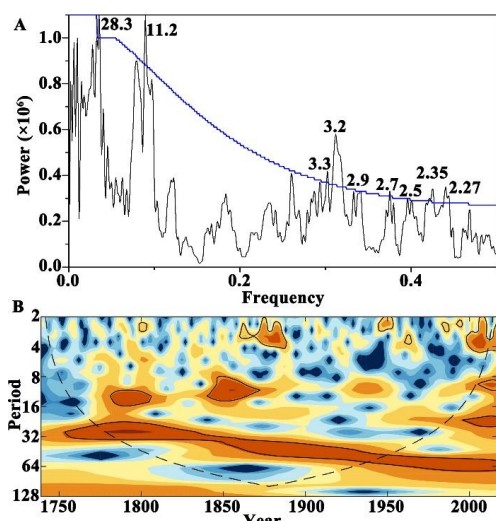


Fig. 8 Power spectral analysis (A) wavelet analysis (B) of the reconstructed PPTp10-c6






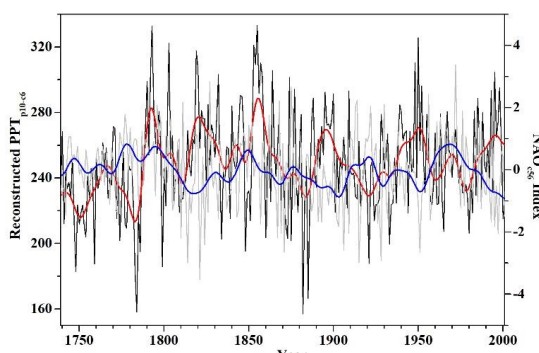


Fig.9 Comparison of the p10-c6 precipitation reconstruction (black line) (red line: 11-year smoothing (FFT filter)) with $NAO_{c56}$ (grey line) (blue line: 11-year smoothing (FFT filter)) (Luterbacher et al., 2002)







## Tables captions



Table 1 Information of the two sampling sites

| Site code | Species | Lat. | Lon. | Elevation | Cores/Trees | Aspect | Slope |
|---|---|---|---|---|---|---|---|
| FL1 | *Pinus koraiensis* | 48.13˚N | 129.18˚E | 440 m | 55/29 | NW | 5 |
| WY1 | *Pinus koraiensis* | 48.2˚N | 129.22˚E | 360 m | 49/24 | W | 10 |







Table 2 Tree-ring width STD chronology statistics

| C/T | MS | Rac | Y/C$_{EPS>0.85}$ | Rar | Rbt | Rwt | SNR | EPS | PC1 |
|---|---|---|---|---|---|---|---|---|---|
| 99/50 | 0.223 | 0.31 | 1748/17 | 0.258 | 0.251 | 0.801 | 30.22 | 0.968 | 42.7% |

Note: C/T the numbers of cores (C) and Trees (T), MS mean sensitivity, Rac first-order
autocorrelation, Y/C$_{EPS>0.85}$ year and minimum number of cores when EPS>0.85, Rar mean inter-
series correlation, Rbt correlation between trees, Rwt correlation within trees, SNR signal-to-noise
ratio, EPS expressed population signal, PC1 % variance explained by the first eigenvector








Table 3 Statistics of calibration and validation results

| Calibration | | | | Validation | | | | | | |
|---|---|---|---|---|---|---|---|---|---|---|
| Period | $R^2$ | $R_{adj}^2$ | F | Period | r | ST | ST1 | t | RE | CE |
| 1956-2017 | 43.9% | 43% | 46.2 | 1956-2017 | 0.63 | 48** | 37 | 2.15 | 0.3966 | |
| 1987-2017 | 53.2% | 52% | 32.9 | 1956-1986 | 0.463 | 21* | 19 | 3.75 | 0.4238 | 0.2083 |
| 1956-1986 | 21.4% | 18.6 | 7.6 | 1987-2017 | 0.729 | 21 | 18 | 2.77 | 0.6285 | 0.5192 |

Note: $R^2$ model explained variance, $R_{adj}^2$ adjusted $R^2$ considering multiple independent variables in
the model, F the F statistic for the statistical significance of the regression models, SE standard error,
r the correlation coefficient of original-reconstructed climate in verification period, ST sign test,
ST1 sign test of the first difference, t the product mean test, RE reduction of error, CE coefficient
of efficiency, *95% confidence level, **99% confidence level








Table 4 Correlations between May-June Asian Polar Vortex Intensity ($APVI_{5-6}$) and Large-Scale Circulation Patterns (concurrent May-June and previous January-February), including El Niño/Southern Oscillation (ENSO) (Trenberth and Stepaniak, 2001; Wu et al., 2003), Multivariate ENSO Index (MEI, Wolter and Timlin, 1998) and Southern Oscillation Index (SOI, Troup, 1965); PDO (Mantua et al. 1997); NAO (Jones et al., 1997) and AO (Zhou et al., 2001)

| $APVI_{5-6}$ | Correlation Coefficient | P |
|---|---|---|
| $ENSO_{1-2}$ | 0.039 | 0.757 |
| $ENSO_{5-6}$ | -0.143 | 0.251 |
| $SOI_{1-2}$ | -0.065 | 0.606 |
| $SOI_{5-6}$ | 0.085 | 0.498 |
| $PDO_{1-2}$ | -0.189 | 0.129 |
| $PDO_{5-6}$ | -0.193 | 0.121 |
| $NAO_{1-2}$ | -0.191 | 0.124 |
| $NAO_{5-6}$ | 0.375 | 0.002 |
| $AO_{1-2}$ | -0.211 | 0.089 |
| $AO_{5-6}$ | 0.255 | 0.039 |