# Peer review of "Northeast China"

_Climate of the Past, 2020_

## Referee Comment (RC1) · Anonymous Referee #1 · 9 Jul 2020

This manuscript uses a large number of tree cores to build a robust tree-ring width chronology. Precipitation, a main limiting factor on tree growth, was reconstructed over the past 270 years. The reconstructed precipitation series was valuable to understand the long-term precipitation variations and its driving factors in the semi-humid Northeast China which produces a large amount of grains in China. In terms of this aspect, this manuscript is useful and valuable. I recommend its publication after some modifications. Major Comments: 1. Since APCI is an important driving factor, it should be introduced in details. For example, how long is the APCI series? How many APCI series are developed? Are they comparable? 2. It is weird to see that the chronology is positively correlated with June minimum temperature, but negatively correlated with

[Figure]

June mean and maximum temperature, even reaching a significant level. Generally, it is thought that minimum, mean, and maximum temperatures change the same way on annual to decadal time scales, at most with some amplitude and/or changing rate differences. The meteorological data should be checked, especially of the June temperature data. Also, can the same results be reached when using data from other nearby meteorological stations? If so, please show them in the supplementary material. Or at least a reasonable explanation should be given why such a weird phenomenon occurs. 3. October-June precipitation was reconstructed. But, each monthly precipitation from previous October to current May is not significant with the chronology (Fig. 4). It is hard to say that they can be represented by the chronology. Maybe only June precipitation is a limiting factor on tree growth here. The chronology has a pretty weak relationship with January-March precipitation, so what is the meaning of the comparison between the reconstructed precipitation with the January-March streamflow in Fig. 7? 4. Paragraph 2 of Possible driving mechanisms. The relationship between the reconstructed precipitation is stronger with May-June APVI index than with previous October to current May APVI index. It is easy to understand the phenomenon when considering that the chronology represents June precipitation, but not October-June precipitation. Therefore, the representative season for reconstruction should be carefully and comprehensively analyzed and decided, not just by the highest correlation between the chronology and climatic factors. 5. As for analyzing the driving mechanisms, the analysis might stop in the APVI based on two reasons. One is that the relationship with NAO index is low and not significant for the period 1748-2001. The other is that their periodicities do not match. Therefore, it is recommended to delete the last paragraph of this part. Minor comments: 6. Paragraph 2 of Introduction. The ms lists a few tree-ring papers from other regions of China here. Introducing the situation of tree-ring studies conducted in Northeast China should follows. 7. L93-95 is unclear. 8. L138-141 is unclear. 9. ENSO is a phenomenon, not an index. So, what indices are used to represent ENSO in Table 4, and relevant content in the ms? 10. How is the growing season defined in the ms? When does the growing season start and end?

11. The correlation coefficients could be provided in Fig. 7 to show the strength of the relationships 12. Units are needed in Fig. 7.

---

## Referee Comment (RC2) · Anonymous Referee #2 · 26 Jul 2020

General Comment: Based on nearly 100 tree cores of Korean pines and climate data between 1956 and 2017, the study reconstructed 270-year precipitation for the northeastern region of China. The manuscript is not very well prepared. Some necessary technical details are missing (e.g., the specifics of the detrending functions, characteristics and the spatial resolution of the APVI), whereas parts of the manuscript are repetitive. Since the intent of the manuscript is to understand soil moisture conditions (drought and wet periods), I recommend that the authors should consider directly reconstructing a drought index based on precipitation data alone, e.g., SPI or SPEI (based on simply Tmax and latitude). A drought index may better serve the purpose of the study. The two indices can be easily derived as the authors already have all the

necessary information.

Specific comments:

1. From Fig. 2, I suspect that the monthly precipitation was only the amount of monthly rainfall. Please clarify whether the precipitation data include the amount of snowfall.

2. For spectral analysis, I would recommend that the authors should consider using multitaper spectral analysis against a red noise background. What does the blue line in Fig. 8A represent? 95% significance level? Also, without information about what different colors represent (power or statistical significance), Fig. 8B by itself is meaningless.

3. The statistics in Table 3 suggest that the regression functions in the 1956-1986 and 1987-2017 periods may have different slopes (temporally unstable). Fig. 5B suggests this may be true. The authors should test the homogeneity of regression slopes to see whether that is the case. I would also like to see a simple sentence states that the residuals of all three regressions met the regular assumptions, given that the residuals indeed met the assumptions.

4. The authors should explain what the black solid and dotted lines represent in Fig. 5C. They should also explain how they derived their 95% CI.

5. The correlations between the reconstructed precipitation and the reconstructed NAO index they used are quite low and statistically not significant. I fail to see the main point of their arguments. Also, please plot the running correlation (e.g., using a 40-year window perhaps) over the common period between the two, so we can see how the correlation evolved. Furthermore, please provide a plausible explanation as to how NAO and the study area's precipitation are correlated.

6. The authors examined the relationships of observed (and reconstructed) precipitation and several large-scale atmospheric-oceanic circulations. After reading the manuscript, however, I still do not have a clear picture of what natural climate drivers

are important to the study area's soil moisture conditions. Another index that the authors may want to look at is the East Asian Winter Monsoon index (use the one developed by Wu and Wang, 2014, An intensity index for the East Asian winter monsoon. J. Clim. 27, 2361–2374.).

Technical comment: Please read through the manuscript again. Remove repetitive information and re-structure the manuscript.

---

## Author Comment (AC1) · 29 Sep 2020

This manuscript uses a large number of tree cores to build a robust tree-ring width chronology. Precipitation, a main limiting factor on tree growth, was reconstructed over the past 270 years. The reconstructed precipitation series was valuable to understand the long-term precipitation variations and its driving factors in the semi-humid Northeast China which produces a large amount of grains in China. In terms of this aspect, this manuscript is useful and valuable. I recommend its publication after some modifications.

Replay: We are very grateful for your comments and constructive suggestions. We have revised the manuscript accordingly and in the following is a point-by-point response to your comments. If we missed anything or if there are things we need to further clarify, please let us know. We will be happy to work with you and the editor to further improve the manuscript as needed.

**Major Comments:**

1. Since APVI is an important driving factor, it should be introduced in details. For example, how long is the APVI series? How many APVI series are developed? Are they comparable?

Reply: We have added more details about the Asian Polar Vortex Intensity Index, including the references on previous studies of circumpolar vortex, computational equation, and the data length. The index used in our study was developed and provided by the National Climate Center of the China Meteorological Administration. The methodology is similar to those described in Burnett (1993), Davis and Benkovic (1992, 1994), and Frauenfeld and Davis (2003).

Burnett, A. W., Size variations and long-wave circulation within the January North Hemisphere circumpolar vortex: 1946–89, J. Clim., 6, 1914–1920, 1993.

Davis, R. E., and S. R. Benkovic, Climatological variations in the Northern Hemisphere circumpolar vortex in January, Theor. Appl. Climatol., 46, 63– 74, 1992.

Davis, R. E., and S. R. Benkovic, Spatial and temporal variations of the January circumpolar vortex over the Northern Hemisphere, Int. J. Climatol., 14, 415– 428, 1994.

Frauenfeld and Davis. 2003. Northern Hemisphere circumpolar vortex trends and climate change implications. JOURNAL OF GEOPHYSICAL RESEARCH, VOL. 108, NO. D14, 4423, doi:10.1029/2002JD002958,

2. It is weird to see that the chronology is positively correlated with June minimum temperature, but negatively correlated with June mean and maximum temperature, even reaching a significant level. Generally, it is thought that minimum, mean, and maximum temperatures change the same way on annual to decadal time scales, at most with some amplitude and/or changing rate differences. The meteorological data should be checked, especially of the June temperature data. Also, can the same results be reached when using data from other nearby meteorological stations? If so, please show them in the supplementary material. Or at least a reasonable explanation should be given why such a weird phenomenon occurs.

Reply: We double-checked the meteorological data, and did not find anything wrong. In addition, we calculated the correlations among TMIN, TMEAN and TMAX, and we found

that the correlation coefficient between June TMIN and TMAX was just 0.31 and it became negative for the first difference data. As we indicated in the Discussion section of the manuscript, the negative correlations of tree growth to early growing-season temperature (mean and maximum), in combination with positive correlations to precipitation, reflect the controlling effect of soil moisture, which is commonly observed arid and semi-arid regions.

3. October-June precipitation was reconstructed. But, each monthly precipitation from previous October to current May is not significant with the chronology (Fig. 4). It is hard to say that they can be represented by the chronology. Maybe only June precipitation is a limiting factor on tree growth here. The chronology has a pretty weak relationship with January-March precipitation, so what is the meaning of the comparison between the reconstructed precipitation with the January-March streamflow in Fig. 7?
Reply: It is true that only June precipitation is strongly correlated with the tree ring series with r = 0.55. With the additional months back to previous October, the correlation increased to 0.663, which means a significant increase in the explained variance of the transfer model for the reconstruction (from 30.3% to 43.9%). In the Lesser Khingan Mountains, snow starts to accumulate in mid-fall and melts after the 17th of April (Zhu et al., 2016). The snow accumulation can insulate the soil and contribute to keeping warm soil temperatures in winter and also improve soil moisture conditions in the following spring. Rapid water absorption by the roots in the following spring and early growing season is the main cause of the positive relationship to tree growth (Fritts 1976). Therefore, while no monthly precipitation from previous October to current May is significantly correlated with the tree-ring chronology (probably because of relatively small and irregular amounts of precipitation in the individual winter and spring months), the total precipitation during these months has important effects on the tree growth in our study area.

Reference: Zhu BB, Man XL, Yu ZX, et al. Forming process of snomelt-runoff of forest watershed in northern region of Da Hinggan Mountains [in Chinese]. Journal of Nanjing Forestry University, 2016, 40(6): 69-75.
Fritts, H. C.: tree rings and climate, Academic Press, London, 1976.
4. Paragraph 2 of Possible driving mechanisms. The relationship between the reconstructed precipitation is stronger with May-June APVI index than with previous October to current May APVI index. It is easy to understand the phenomenon when considering that the chronology represents June precipitation, but not October-June precipitation. Therefore, the representative season for reconstruction should be carefully and comprehensively analyzed and decided, not just by the highest correlation between the chronology and climatic factors.
Reply: Although the PPTp10-c6 is the limiting factor of radial growth of trees, more than 60% of the observed PPTp10-c6 occurs in May and June, explaining more than 71% of the total variance in PPTp10-c6. Therefore, precipitation in May and June has an important role on tree growth in our study area. However, this does not mean that the precipitation from previous October to current April has no effects on the tree-growth, as we explained above. We also investigated the relationship between the APVI and precipitation in these months, but we did not find any statistically significant correlations. This is probably resulted from relatively small amount of precipitation in winter and spring months. In other words, the

circumpolar vortex may be inducive to surface cyclonic activities in winter and spring, but the main rainy belt is located in central and southern China during these seasons.

[Figure]

**Fig. 3.** A composite map of rain belts advance in China during the major rainy seasons (Unit: pentad). The thin lines denote the rough locations of high value areas of standardized rainfall amounts, while the thick lines denote the route of the advance of rain belts. The square denotes the central region of anticlockwise rotation of rain belts in China

Ding, Yihui and Wang, Zunya. 2008. A study of rainy seasons in China. *Meteorology and Atmospheric Physics*, 100, 121-138

5. As for analyzing the driving mechanisms, the analysis might stop in the APVI based on two reasons. One is that the relationship with NAO index is low and not significant for the period 1748-2001. The other is that their periodicities do not match. Therefore, it is recommended to delete the last paragraph of this part.
Reply: Thank you very much and we agree to your suggestion. We deleted the last paragraph of the part and related contents.
**Minor comments:**
6. Paragraph 2 of Introduction. The ms lists a few tree-ring papers from other regions of China here. Introducing the situation of tree-ring studies conducted in Northeast China should follows.
Reply: We have added a few more references in the manuscript accordingly. In addition, we have added more content about tree-ring studies in our study area in Lines 53-54 on page 3.

7. L93-95 is unclear.
Reply: We have revised it as recommended. Please refer to Lines 106-108 in the revised manuscript.
8. L138-141 is unclear.
Reply: We now clarified the definition according to the agency's source, added the equation of its calculation, and additional descriptions and references about this index in Lines 176-214.
9. ENSO is a phenomenon, not an index. So, what indices are used to represent ENSO in Table 4, and relevant content in the ms?
Reply: we have deleted content about ENSO. We used MEI and SOI to measure the phase

and intensity of the ENSO events.

10. How is the growing season defined in the ms? When does the growing season start and end?

Reply: The growing season is defined when the day mean temperature is more than 5 ℃ for continual 5 days (Frich et. al., 2002). According to the definition, we have clarified the months of the growing season as recommended (Line 84).

Frich, P., Alexander, L. V., Della-Marta, P., et.al.: Observedcoherent changes in climatic extremes during the second half of the twentieth centure. Climate Research, 19, 193-212, 2002.

Frich et al. 2002. Observed coherent changes in climatic extremes during the second half of the twentieth century. Climate Research Vol. 19: 193–212.

11. The correlation coefficients could be provided in Fig. 7 to show the strength of the relationships.

Reply: We calculated the correlation coefficients between our reconstructed series with other series, the correlation coefficient is 0.1 between our reconstructed series and the PDSI series in the Greater Khingan Mountains, and 0.21 between our reconstructed series and precipitation in Northeastern China. Although the correlations are low, they are in good agreement in certain time periods. Therefore, we did not add the correlation coefficients in Fig. 7.

12. Units are needed in Fig. 7.

Reply: We have added the units as recommended.

---

## Author Comment (AC2) · 29 Sep 2020

General Comment: Based on nearly 100 tree cores of Korean pines and climate data between 1956 and 2017, the study reconstructed 270-year precipitation for the northeastern region of China. The manuscript is not very well prepared. Some necessary technical details are missing (e.g., the specifics of the detrending functions, characteristics and the spatial resolution of the APVI), whereas parts of the manuscript are repetitive. Since the intent of the manuscript is to understand soil moisture conditions (drought and wet periods), I recommend that the authors should consider directly reconstructing a drought index based on precipitation data alone, e.g., SPI or SPEI (based on simply Tmax and latitude). A drought index may better serve the purpose of the study. The two indices can be easily derived as the authors already have all the necessary information.

Reply: We are very grateful for your constructive comments and suggestions. We have carefully revised the manuscript according to your suggestions. If we missed anything or if there are things we need to further clarify, please let us know. We will be happy to work with you and the editor to further improve the manuscript as needed.

According to your suggestion, we have added more details regarding the data and methods, including more details about the polar vortex intensity index in Lines 176-214, as well as the detrending method of the tree-ring chronology construction in Lines 145-148. In the Discussion section, we also presented the map to show the relationship between our reconstruction and the SPEI (Fig. 6A). As you suggested, there is a strong relationship with SPEI. However, the reconstruction of precipitation provides the advantage of examining a single climatic factor, which made it easier for us to explain the mechanisms of the interannual variation patterns. In the Lesser Khingan Mountains, snow accumulates in mid-fall and winter and melts after the 17th of April (Zhu et al., 2016). The snow accumulation can insulate the soil and contribute to keeping warm soil temperatures in winter. Additionally, rapid water absorption by the roots in the following spring and early growing season contributes to enhanced growth rates (Fritts 1976?). Therefore, while no monthly precipitation from previous October to current May is significantly correlated with the tree-ring chronology (probably because of relatively small and irregular amounts of precipitation in the individual winter and spring months), the total precipitation during these months has important effects on the tree-growth in our study area. Therefore, we decided to reconstructing the PPTp10-c6.

Specific comments:

1. From Fig. 2, I suspect that the monthly precipitation was only the amount of monthly rainfall. Please clarify whether the precipitation data include the amount of snowfall.

Reply: The precipitation data from the meteorological station include both snowfall (as water equivalent) and rainfall and we have clarified this in the manuscript (Line 164).

2. For spectral analysis, I would recommend that the authors should consider using multitaper spectral analysis against a red noise background. What does the blue line in Fig. 8A represent? 95% significance level? Also, without information about what different colors represent (power or statistical significance), Fig. 8B by itself is meaningless.

Reply: We did spectral analysis using multitaper against a red noise background using matlab software. We have changed the Fig. 8 and clarified the different lines representing the

confidence levels.

3. The statistics in Table 3 suggest that the regression functions in the 1956-1986 and 1987-2017 periods may have different slopes (temporally unstable). Fig. 5B suggests this may be true. The authors should test the homogeneity of regression slopes to see whether that is the case. I would also like to see a simple sentence states that the residuals of all three regressions met the regular assumptions, given that the residuals indeed met the assumptions.

Reply: Fig. 5B shows the observed and reconstructed p10-c6 precipitation (PPTp10-c6) for the full calibration period 1956-2017. Different slopes may exist for the regression lines, but the purpose of split validation is to mainly make sure that the relationship is statistically significant without large biases for the two independent time periods. Of course, the optimal results should be consistent regression models. What is important is that the model based on the entire calibration period was robust as this one was used for the reconstruction. The results of t-test indicated that the regression functions in the 1956-1986 and 1987-2017 periods have same slopes. The residuals of all three regression models met the regular assumptions of regression: the residuals are all tested as random variables; and the histograms below indicated that the residuals were likely to be normally distribution (Fig.1). We have added the statement regarding the residuals and assumptions of regression analysis in the revised manuscript (Lines 340-343).

[Figure]

Fig.1 the scatter plots between residuals and fitted values, and histograms of residuals for different validation periods.

4. The authors should explain what the black solid and dotted lines represent in Fig. 5C. They

should also explain how they derived their 95% CI.

Reply: We have added the explanations about the black solid and dotted lines in the graph title. Since the calculation of CI of regression model estimates is a rather standard procedure in regression analysis, we would like to refer to a textbook of statistical methods as the reference in the figure caption (Ott, 1988, p. 356).

Ott, L. 1988. An Introduction to Statistical Methods and Data Analysis, 3rd ed. PWS-Kent Publishing Co. Boston, USA. 835 pp.

5. The correlations between the reconstructed precipitation and the reconstructed NAO index they used are quite low and statistically not significant. I fail to see the main point of their arguments. Also, please plot the running correlation (e.g., using a 40-year window perhaps) over the common period between the two, so we can see how the correlation evolved. Furthermore, please provide a plausible explanation as to how NAO and the study area's precipitation are correlated.

Reply: We originally included NAO as a possible factor of regional circulation of our study region because of its broad impact across the northern Eurasia continent. Our results showed that May-June NAO is positively correlated with May-June APVI, while the latter is negatively correlated to May-June precipitation. We added the explanation to the connection between APVI and precipitation in the revised manuscript (Lines 499-503). Because the mechanisms of how NAO influences weather conditions in northern Eurasia are complex and not fully understood (Schlichtholz 2019) and its correlation with precipitation is low, as you pointed out, we now have deleted the related content.

Schlichtholz, P. 2019. https://www.nature.com/articles/s41598-019-51019-w

6. The authors examined the relationships of observed (and reconstructed) precipitation and several large-scale atmospheric-oceanic circulations. After reading the manuscript, however, I still do not have a clear picture of what natural climate drivers are important to the study area's soil moisture conditions. Another index that the authors may want to look at is the East Asian Winter Monsoon index (use the one developed by Wu and Wang, 2014, An intensity index for the East Asian winter monsoon. J. Clim. 27, 2361–2374.).

Reply: Since NAO, PDO and ENSO did not show strong correlations, we have decided to delete the discussion about these large-scale atmospheric-oceanic circulations, and just focus on the correlation between the APVI and precipitation in our study area. According to your suggestion, we calculated the correlation between precipitation and the EAWMI, but the correlation coefficient is low and not statistically significant, r = -0.121 (p=0.174). Apparently, the East Asian Winter Monsoon is more of a factor of winter and spring temperature, and does not have a significance effect on precipitation in our study area.

Technical comment: Please read through the manuscript again. Remove repetitive information and re-structure the manuscript.

Reply: We made editorial changes throughout the manuscript to improve language usage and removed repetitive information.